# Posterior Preventive Foraminotomy before Laminectomy Combined with Pedicle Screw Fixation May Decrease the Incidence of C5 Palsy in Complex Cervical Spine Surgery in Patients with Severe Myeloradiculopathy

**DOI:** 10.3390/jcm12062227

**Published:** 2023-03-13

**Authors:** Yong-Ho Lee, Mahmoud Abdou, Ji-Won Kwon, Kyung-Soo Suk, Seong-Hwan Moon, You-Gun Won, Tae-Jin Lee, Byung-Ho Lee

**Affiliations:** 1Department of Orthopedic Surgery, Yonsei University College of Medicine, Seoul 04799, Republic of Korea; 2Department of Orthopedic Surgery, Yonsei Baro Chuk Hospital, Seoul 06273, Republic of Korea; 3Department of Orthopedic Surgery, Fayoum University College of Medicine, Fayoum 2933051, Egypt

**Keywords:** C5 palsy, foraminotomy, uncovertebrectomy, laminectomy fusion, cervical myelopathy, myeloradiculopathy

## Abstract

C5 palsy is a frequent sequela of cervical decompression surgeries for cervical myeloradiculopathy. Although many researchers have suggested various risk factors, such as cord shifting and the correction of lordotic angles, the tethering of the C5 root beneath the narrow foramen is an independent risk factor for C5 palsy. In this study, we tried to investigate different techniques for foramen decompression with posterior cervical fusion and assess the incidence of C5 palsy with each technique depending on the order of foraminal decompression. A combined 540° approach with LMS and uncovertebrectomy was used in group 1. Group 2 combined a 540° approach with pedicle screws and posterior foraminotomy, while posterior approach only with pedicle screws and foraminotomy was used in group 3. For groups 2 and 3, prophylactic posterior foraminotomy was performed before laminectomy. Motor manual testing to assess C5 palsy, the Neck Disability Index (NDI) and the Japanese Orthopedic Association (JOA) scores were determined before and after surgery. Simple radiographs, MRI and CT scans, were obtained to assess radiologic parameters preoperatively and postoperatively. A total of 362 patients were enrolled in this study: 208 in group 1, 72 in group 2, and 82 in group 3. The mean age was 63.2, 65.5, and 66.6 years in groups 1, 2, and 3, respectively. The median for fused levels was 4 for the three groups. There was no significant difference between groups regarding the number of fused levels. Weight, height, comorbidities, and diagnosis were not significantly different between groups. Preoperative JOA scores were similar between groups (*p* = 0.256), whereas the preoperative NDI score was significantly higher in group 3 than in group 2 (*p* = 0.040). Mean JOA score at 12-month follow-up was 15.5 ± 1.89, 16.1 ± 1.48, and 16.1 ± 1.48 for groups 1, 2, and 3, respectively; it was higher in group 3 compared with group 1 (*p* = 0.008) and in group 2 compared with group 1 (*p* = 0.024). NDI score at 12 months was 13, 12, and 13 in groups 1, 2, and 3, respectively; it was significantly better in group 3 than in group 1 (*p* = 0.040), but there were no other significant differences between groups. The incidence of C5 palsy was significantly lower in posterior foraminotomy groups with pedicle screws (groups 2 and 3) than in LMS with uncovertebrectomy (group 1) (*p* < 0.001). Thus, preventive expansive foraminotomy before decompressive laminectomy is able to significantly decrease the root tethering by stenotic lesion, and subsequently, decrease the incidence of C5 palsy associated with posterior only or combined posterior and anterior cervical fusion surgeries. Additionally, such expansive foraminotomy might be appropriate with pedicle screw insertion based on biomechanical considerations.

## 1. Introduction

Cervical myeloradiculopathy is the most frequent cause of spinal cord impairment in the elderly population [1,2], leading to an increase in the frequency of cervical decompressive surgery. Postoperative C5 palsy represents one of the commonest complications, occurring in up to 28.9%, mainly with fusion surgeries [3,4,5,6], and is almost always associated with severe morbidity and reduced quality of life [7].

There is no consensus about the exact cause of postoperative C5 palsy, but several hypotheses have been proposed [8]. One hypothesis suggests that it is due to dural shift with wide laminectomy, leading to the tethering of the C5 root beneath the narrow foramen [9,10]. In contrast, another study suggests that it is caused by iatrogenic foraminal stenosis during cervical lordosis correction [11]. Some authors believe it is related to symptom duration or a long period of subclinical disease before intervention [12]. Many studies confirmed that preoperative foramen size is the most significant independent risk factor for C5 palsy [4,13], but others assume it results from iatrogenic injury during decompression [14].

As preoperative foramen size plays a vital role in the development of C5 palsy according to studies in the literature, there have been no studies related to the order of foraminotomy and laminectomy, which can affect the C5 root tethering and stretch.

Therefore, we aimed to conduct this study and compare the incidence and recovery rates of C5 palsy after three different decompression and fusion strategies, including the order of foraminotomy and laminectomy performed by either foraminotomy or uncovertebrectomy.

## 2. Materials and Methods

This study was approved by our institutional review board and ethics committee (IRB no. 4-2021-1469). This retrospective analysis of prospectively collected data included patients with cervical spine myeloradiculopathy who underwent surgical decompression at our hospital from April 2014 to January 2021. Surgical decompression was indicated for patients with severe myeloradiculopathy (with a Japanese Orthopedic Association (JOA) score < 12). At the same time, conservative management (anti-inflammatory drugs, cervical immobilization in neck collar, physiotherapy, activity modification, pain block, and neuroplasty) was initiated for patients with mild to moderate symptoms (JOA ≥ 12) [15]. If symptoms did not improve or deteriorate with conservative treatment, surgical intervention was suggested [16]. Patients with tumors, traumatic lesions, infectious diseases, or with no C4–5 foramen stenosis were excluded from this study.

A combined anterior and posterior approach was used for patients with less lordosis associated with severe myeloradiculopathy symptoms and severe foraminal stenosis with decreased width and height based on the suggested algorithm. The anterior and posterior combined approach was performed only in cases with a k-line negative and rigid kyphosis in lesions of 3 or more stages [17]. The severity of foraminal stenosis could be graded as levels 1–2 of foraminal stenosis using the MRI Grading System for cervical foraminal stenosis from axial scan; grade 0 = absence of foraminal stenosis with the narrowest width of the neural foramen being greater than the width of the extraforaminal nerve root (EFNR); grade 1 = the narrowest width of the neural foramen equal to or greater than 50% of the width of the EFNR; grade 2 = the narrowest width of the neural foramen equal to or less than 50% of the width of the EFNR [18,19]; or grade 3 = based on the lumbar foraminal stenosis grade system from sagittal scan with perineural fat loss and decreased foraminal height [20].

To achieve better foramen decompression, direct foraminotomy or uncovertebrectomy, along with indirect decompression by increasing foraminal height through an allospacer insertion, were added during anterior surgery, and additional effective kyphosis correction was also achieved [17].

A posterior approach alone was used for patients with neutral or lordosis in radiological parameters foraminal stenosis graded 1–2 with a relatively well-preserved foraminal height that was adequately decompressed by laminectomy in addition to expansive posterior foraminotomy with pedicle screws [17,18,19].

The patients were divided into three groups: (1) combined anterior and posterior approaches with lateral mass screws (LMS) and uncovertebrectomy (group 1: PAP LMS group); combined anterior and posterior approaches with pedicle screws and foraminotomy (group 2: PAP pedicle group); and a posterior approach with pedicle screws and foraminotomy (group 3: posterior pedicle group). Notably, foraminotomy was performed for all foraminal stenosis levels before laminectomy to lessen the stretch of C5 roots in groups 2 and 3.

We intended to begin with a posterior rather than an anterior approach to allow for more uncomplicated posterior dissection and neurological decompression, as well as hemostasis and pedicle screw insertions. Then, ACDF using allospacers with foraminotomy and uncovertebrectomy was performed, followed by posterior rod assembly [21,22]. 

Group 1 patients underwent posterior laminectomy and fusion but no posterior foramintomy was performed; multilevel laminectomy and LMS insertion were performed from C3 to C7 [23] (Figure 1a). One week later, anterior cervical decompression and fusion with anterior uncovertebrectomy (Figure 2 and Figure 3) were performed, followed by posterior rod insertion and posterolateral fusion with autobone graft during the same session [24,25].

Groups 2 and 3 patients first underwent a posterior cervical approach (Figure 1b,c) with pedicle screw fixation used whenever the pedicle diameter permitted. Our novel free-hand technique (medial pedicle pivot point (MPPP) technique) was used for pedicle screw insertion [22]. If the pedicle was <3.5 mm or a vertebral artery anomaly was present, we avoided pedicle screws and used LMS instead [22,26,27]. Then, bilateral microscopic foraminotomy was performed using a 3 mm burr to initiate the foraminotomy before laminectomy [28]. A small portion of the inferior articular process and part of the superior articular process were drilled until a defect was created over the foramen. Bone removal was then continued using a 1 mm Kerrison punch or 1–2 mm sized cervical curette until the microhook moved freely above the nerve root [29] (Figure 3 and Figure 4). Most of the facet joint and lateral mass were removed. As a result, all of the foramen was exposed, from cephalad to caudal pedicles. Next, decompressive laminectomy was performed for central decompression, and the removed bone was cut into small pieces for a posterolateral bone graft. Group 2 patients underwent anterior cervical discectomy with additional anterior foraminotomy and fusion in the following week, using allospacers to increase forminal height and inserting a posterior rod assembly in the same session [25].

All patients underwent neurologic function monitoring intraoperatively via transcranial motor evoked potential (MEPs). This was mostly performed for the detection of deterioration of myelopathy symptoms, but according to Yanase et al., intraoperative wave changes in transcranial MEPs are predictive of postoperative C5 palsy [31]; however, there were no significant changes in MEP with patients who developed C5 palsy.

All patients underwent motor manual testing (grading muscle strength from 0 to 5) preoperatively and immediately postoperatively, following each surgical intervention [10]. The motor check was carried out by different two orthopedic surgeons. The decrease by at least one degree of deltoid muscle function has been used to define C5 palsy [10].

All patients also underwent repeat testing within the first 6 postoperative weeks. Neck Disability Index (NDI) and JOA scores for myelopathy were obtained preoperatively and at 6 and 12 postoperative months. Cervical lordosis was assessed by measuring the C2–C7 Cobb angle, C2–C7 sagittal vertical axis (SVA), C2 slope, T1 slope, and T1s–CL (T1 slope minus cervical lordosis (C2–C7 Cobb angle)) on lateral radiographs preoperatively and postoperatively. Finally, all cases with pedicle screws had a postoperative CT to ensure screw position was correct.

## 3. Results

A total of 362 patients were enrolled in this study: 208 in group 1 (PAP with LMS), 72 in group 2 (PAP with pedicle screws), and 82 in group 3 (posterior only with pedicle screws) (Table 1).

Group 1 was relatively larger than other groups due to the later introduction of pedicle screws in our institute and the limitation of the use of pedicle screws when the pedicle axial diameter was less than 3.5 mm. So, we used the propensity score to select matched groups for our study to obtain significant results. Cases presented with myeloradiculopathy symptoms of C4–5 foramen stenosis were selected. Females composed 25.5% of patients in group 1, 43.1% of patients in group 2, and 22% of patients in group 3.

The incidence of C5 palsy was significantly lower in posterior preventive foraminotomy groups with pedicle screws (groups 2 and 3) than LMS group with uncovertebrectomy (group 1) (*p* < 0.001). It was (10.1%) in group 1, 4.2% in group 2, and 0% in group 3. C5 palsy was observed only in groups 1 and 2, and all palsies in group 1 occurred in the interval between the posterior and anterior procedures. All cases recovered within 3–6 postoperative months.

The preoperative C2–C7 Cobb angles were 7, 5.5, and 9 degrees in groups 1–3, respectively; they were significantly higher in group 3 than in group 2 (*p* = 0.029), but there were no other significant differences between groups. Preoperative C2–C7 SVA was 19, 21, and 19.5 mm in groups 1–3, respectively, with no significant differences between groups (Table 2).

Postoperatively, the mean C2–C7 Cobb angle was 18, 26, and 13 degrees in groups 1–3, respectively; the difference between group 2 and the two other groups was highly significant (*p* < 0.001 for both) (Table 3). Postoperative C2–C7 SVA was 23.5, 21.5, and 22 mm in groups 1–3, respectively; there was no significant difference between groups (*p* = 0.322). Postoperative T1s–CL was 13, 2, and 12 degrees for groups 1–3, respectively; it was significantly better in group 2 than in groups 1 and 3 (*p* < 0.001 for both), but there were no other significant differences between groups. To summarize, cervical lordosis was better corrected in groups 1 and 2 than in group 3 and was best corrected in group 2.

Preoperative JOA scores were similar between groups (*p* = 0.256), whereas the preoperative NDI score was significantly higher in group 3 than in group 2 (*p* = 0.040) (Table 4). Preoperative NDI scores were 20, 19, and 20 in groups 1–3, respectively. At 12 months, NDI (*p* = 0.046 and JOA (*p* = 0.011) scores differed significantly between groups. Mean JOA score at 12-month follow-up was 15.5 ± 1.89, 16.1 ± 1.48, and 16.1 ± 1.48 for groups 1, 2, and 3, respectively; it was higher in group 3 compared with group 1 (*p* = 0.008) and in group 2 compared with group 1 (*p* = 0.024). NDI score at 12 months was 13, 12, and 13 in groups 1, 2, and 3, respectively; it was significantly better in group 3 than in group 1 (*p* = 0.040), but there were no other significant differences between groups.

## 4. Discussion

Surgical intervention is suggested for patients with severe or progressive cervical myeloradiculopathy symptoms [32]. C5 palsy is a frequent complication of cervical decompression surgery, with a reported incidence of up to 28.9% [3,4,5].

According to Shurab Rachmani et al., ligamentum flavum does not enter the foramen, unlike the lumbar spine. Therefore, it is exposed to the risk of thermal damage during manipulation around the foramen. Takenaga et al. reported lower C5 palsy in the group using chilled irrigation than in the control group in 400 cases of open door laminoplasty [33,34]. In all cases of this study, during manipulation around the foramen, in order to reduce the possibility of thermal injury that may occur during drilling, saline irrigation was performed.

Posterior cervical decompression (laminectomy or laminoplasty) is an effective treatment for myeloradiculopathy [35]. When performed alone, decompression surgery has a relatively low incidence of C5 palsy [36]. However, laminectomy and laminoplasty are less effective than a fusion for improving myeloradiculopathy symptoms and neck pain/disability, especially in patients with deformity or instability or severe foraminal stenosis [27,37,38,39].

Thus, it is necessary to identify the best strategy to resolve myeloradiculopathy symptoms and prevent C5 palsy.

Bilateral prophylactic foraminotomy [40] has been recommended to decrease the risk of C5 palsy associated with posterior decompression procedures. In their literature review, Pan et al. [3] noted that foraminotomy was a significant independent factor in reducing postoperative C5 palsy. Similarly, Imagama et al. [41] recommended foraminotomy based on the results of their large, multicenter study. Nevertheless, studies documenting a lower incidence of C5 palsy (0.6%) after foraminotomy involved only non-fusion procedures [36]. To our knowledge, only one study examined the incidence of C5 palsy with laminectomy and fusion using LMS in association with posterior foraminotomy [42]. In that study, the C5 palsy incidence was higher (14.1%) [42] and similar to the rates previously reported for non-foraminotomy fusion procedures [3,6]. This may have been due to insufficient foraminotomy to preserve sufficient bone for LMS. [30] The incidences in our foraminotomy groups (groups 2 and 3) were much lower (4.2% and 0%, respectively), suggesting that effective foraminotomy is generally not enough for LMS.

Pedicle screws provide a better opportunity for sufficient aggressive foraminotomy with their anatomical advantage because of a superior entry point in comparison to LMS that allows the safe removal of most of the facet joint. We can expose the whole foramen from the cephalad to caudal pedicle screws [22]. In addition to a biomechanical advantage, compared with LMS, pedicle screw use allows aggressive foraminotomy without reducing stability. Additionally, pedicle screws are much better than LMS in preventing cage subsidence with time, which preserves foramen height and maintains cervical lordosis when a combined approach is used [30]. Despite the previous advantage, pedicle screws are not widely used because of fears of neurovascular complications, especially with the lack of navigation systems in most institutes. However, pedicle screws have become popular at our institute thanks to our novel free-hand technique (MPPP) that does not require a navigation system and no documented complications were observed in our previous cohort [22]. Additionally, the three cases who developed C5 palsy had no abnormal postoperative CT findings.

When using LMS, we initially added uncovertebrectomy to provide better foramen decompression, but patients who underwent posterior foraminotomy group 1, as in groups 2 and 3, had a lower C5 palsy rate than those who underwent uncovertebrectomy (10.1% in group 1 versus 4.2% in group 2). This may have been because of the delay in foramen decompression by uncovertebrectomy, as all cases of C5 palsy occurred during the interval between approaches in the LMS group. Owing to this, we were also keen on performing foraminotomy in pedicle groups even before the laminectomy out of concern that a delayed procedure might not be able to stop the abrupt tethering of the C5 root caused by the sudden dural shift after laminectomy. Thus, we recommend performing foraminotomy before laminectomy, which causes the root to run momentarily before the abrupt dural shift. In addition, this will make the procedure technically easier due to less bleeding in the surgical field. Of course, it is possible to combine posterior foraminotomy with LMS; however, the most fixation stability and the most stressed area in LMS is the inferomedial part of the entry point according to our previous biomechanical study, so we were sure that such aggressive foraminotomy would lead to weak bone purchase with LMS [30].

Collectively, foramen decompression by either uncovertebrectomy or foraminotomy resulted in substantially lower C5 palsy rates (10.1% in group 1, 4.2% in group 2, and 0% in group 3) than previously reported in laminectomy fusion studies. In the meta-analysis by Wang et al., the incidence after posterior fusion was 12.2% overall, including 13.3% in patients with cervical spondylotic myelopathy and 13.1% in those with the ossification of the posterior longitudinal ligament [6,43]. The incidence of C5 palsy was even higher in some studies: 28.9% after posterior laminectomy and fusion in the study reported by Blizzard et al. [3,44] and 36.4% after combination surgery in the study by Chang et al. [45]. Although our C5 palsy rates were low, we surmise that the occurrence of this complication may be related to the severity of the foraminal stenosis or the duration of preoperative symptoms, as we used a combined approach in patients with severe stenosis to preserve the foramen height for better foramen decompression, as has been performed in previous studies [42,46]. Thus, it may be impossible to eliminate C5 palsy. Of note, a few authors have recommended against foramen decompression [47]. For example, Cho et al. suggested that foramen decompression leads to the excessive manipulation of nerve roots [48]. However, our experience indicates that microscopic decompression with a 3 mm burr and small 1 mm curette or Kerrison punch is relatively safe and effective.

We also examined which type of strategy led to the most improvement in myelopathy symptoms, neck pain, and disability. It is well known that fusion surgery provides better outcomes for myelopathy and neck disability symptoms [27,49]. However, there is no consensus regarding the optimal degree of lordosis correction, although when postoperative cervical alignment is near neutral, it is assumed that this is a better outcome [50,51]. The preoperative C2–C7 Cobb angle was significantly better in group 3 than in the other groups, as we reserved the combined approaches for patients with severe stenosis and deformity. The preoperative C2 slope was significantly higher in group 2 than in groups 1 and 3, indicating that group 2 required increased surgical correction. The postoperative correction of C2–C7 lordosis was significantly greater in group 2 than in the other groups, emphasizing the tremendous corrective power of combined approaches, mainly when accomplished with pedicle screws. The delta C2 slope angle also showed significantly greater correction in groups 1 and 2 than in group 3. 

The postoperative C2–C7 Cobb angle values were 18, 26, and 13 degrees in our three groups, which were very close to the average values of 15–25 degrees reported by Gore et al. and Blondel et al. [52,53]. C2–C7 SVA values were 23.5, 21.5, and 22 mm in our three groups, which were also similar to the average value (22.4 mm) reported by Blondel et al. [53]. Of note, an SVA >40 mm is associated with high levels of disability [54]. These parameters showed preserved cervical alignment in all three groups, although the C2–C7 Cobb angle was better with the combined posterior and anterior approaches, especially in group 2. As a consequence, myelopathy symptoms improved in all groups, consistent with the results of previous studies [27,49].

A problem with kyphosis correction is that it may increase the dural shift, in addition to decreasing the foramen height, and thereby increase tension on the C5 root [6,7,10]. However, this did not seem to occur in our patients, as group 2 underwent significant kyphosis correction but had a lower C5 palsy rate compared to group 1, suggesting that C5 palsy is the result of not only dural shift but also the stretch of C5 root from foraminal stenosis. Therefore, we anticipate that preventive expansive foraminotomy before laminectomy will result in a lower incidence of C5 palsy with better cervical anatomic parameters.

Although conservative treatment is usually successful for treating C5 palsy, and all cases recovered within a few months, with no difference in time to recovery between groups [7,10], recovery may be prolonged or may not even occur in some patients [55]. As C5 palsy remains a burden on the economy and the patient’s quality of life, more research is warranted to decrease its incidence [7].

One of the major concerns of the study design is how to control and classify the patients who would undergo PAP surgery and posterior-only surgery based on the severity of central and foraminal stenosis. This, however, is relevant to the unique healthcare system of the authors’ country.

In South Korea, the Health Insurance Review and Assessment Service (HIRA) reviews and assesses healthcare costs and healthcare service quality, in addition to supporting the national health insurance policy in determining medical fee categories and drug prices. Ministry of Health and Welfare (MoHW) oversees the national health insurance system. There are two other institutions, the National Health Insurance Service (NHIS), which serves as the insurer, and the Health Insurance Review and Assessment Service (HIRA), which conducts reviews and the assessment of medical fees. The reimbursement process starts with the health institution filing a claim for medical fees to HIRA. After HIRA reviews the claim, it notifies the NHIS of the results and contacts the health institutions regarding payment. The “Fee-for-Service (FFS)” is the traditional reimbursement system in South Korea. In order to reduce the number of unnecessary services, the payment standard is very strictly controlled [56].

If PAP surgery was carried out for severe myeloradiculopathy but the reviewers at the HIRA do not agree with the PAP surgery, then the fee for one of posterior or anterior surgery would be cancelled and not be paid to the hospital and surgeons. This procedure could control the standardized surgical decision for the patients with severe myeloradiculopathy, and those with similar severity of diseases could be treated with the same surgical procedures.

## 5. Conclusions

Although C5 palsy after cervical decompression surgery cannot be avoided entirely, particularly in patients with severe foraminal stenosis, preventive expansive foraminotomy before decompressive laminectomy can significantly decrease root tethering by stenotic lesion and, as a result, decrease the incidence of C5 palsy associated with posterior and combined posterior and anterior cervical fusion surgeries. Notably, wider bilateral foraminotomy could be performed when using pedicle screws compared to using LMS, based on biomechanical considerations. This is the first case series of preventive foraminotomy before laminectomy with a posterior cervical fusion using pedicle screws, and additional prospective studies are required to further assess this technique.

## Figures and Tables

**Figure 1 jcm-12-02227-f001:**
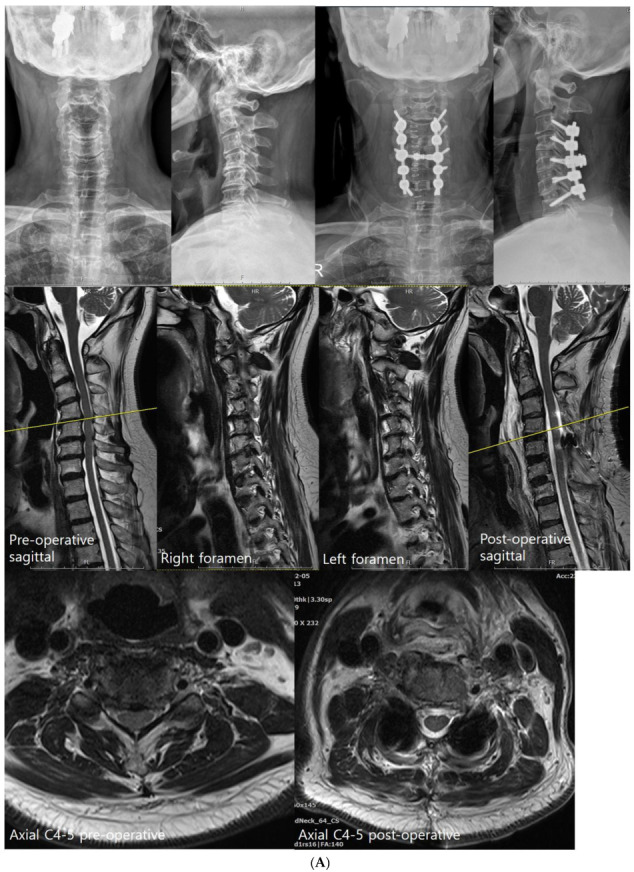
(**A**) Combined posterior and anterior approach with lateral mass screws (LMS) at C3–C7 and pedicle screws at C7 (Group 1: PAP LMS with uncovertebrectomy). (**B**) Combined posterior and anterior approach with pedicle screw fixation at C3–C7 and multiple foraminotomies (Group 2: PAP pedicle with foraminotomy). Both groups 1 and 2 demonstrated severe foraminal stenosis with decreased foraminal height, which was resolved through additional ACDF surgery. (**C**) Posterior approach with pedicle screws at C3–C7 and multiple foraminotomies (Group 3: posterior pedicle with foraminotomy). Compared with groups 1 and 2, group 3 demonstrated relatively well-preserved foraminal height, although with foraminal stenosis, which could be decompressed successfully by posterior foraminotomy only.

**Figure 2 jcm-12-02227-f002:**
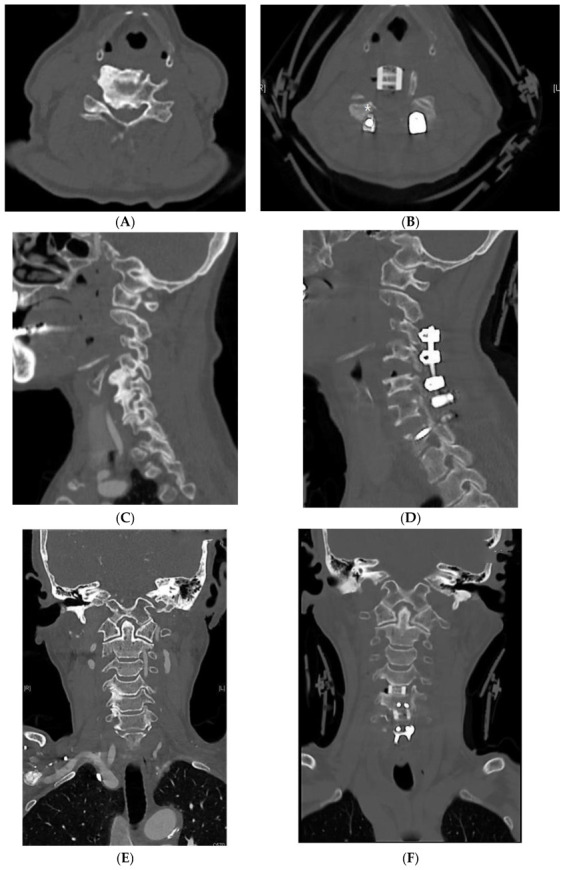
PAP LMS with uncovertebrectomy technique. (**A**) Preoperative CT axial image showing foraminal stenosis. (**B**) Postoperative axial image showing uncovertebrectomy with LMS and allospacer. Depending on the amount of dural shift, some risk of tethering of the C5 root still remains (marked with *). (**C**) Preoperative CT sagittal image showing severe C4–C5 foraminal stenosis. (**D**) Postoperative CT sagittal image showing the extent of foramen height, in addition to an uncovertebrectomy in the presence of the allospacer and LMS. (**E**) Preoperative CT coronal image showing foraminal stenosis. (**F**) Postoperative CT coronal image showing uncovertebrectomy. CT, computed tomography; LMS, lateral mass screws; PAP, posterior anterior–posterior.

**Figure 3 jcm-12-02227-f003:**
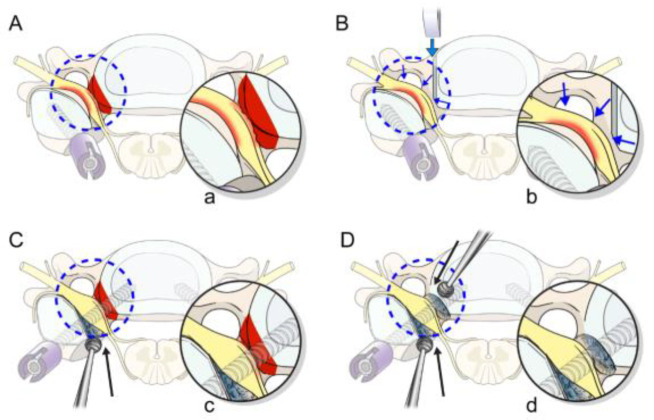
Illustration showing the effects of posterior decompression alone, compared with other techniques of foramen decompression. (**A**) Laminectomy alone with lateral mass screws shows tethering of the C5 root with sudden dural shift inside the narrow foramen. (**B**) Uncovertebrectomy with lateral mass screws. Although the effective decompression of the C5 root is achieved, this technique does not eliminate the possibility of C5 root impingement because the lateral mass is preserved for better screw purchase. (**C**) Posterior foraminotomy and laminectomy with pedicle screws. Posterior foraminotomy was carried out before laminectomy, which offers the least chance of the impingement of the C5 root after the dural posterior shift with laminectomy. However, the uncinate process still compresses the root, especially when foraminal stenosis has severely decreased foraminal height. (**D**) Adding the anterior decompression and insertion of allospacers to maximize the size of the foramen after posterior foraminotomy in the PAP pedicle group. PAP, posterior anterior–posterior.

**Figure 4 jcm-12-02227-f004:**
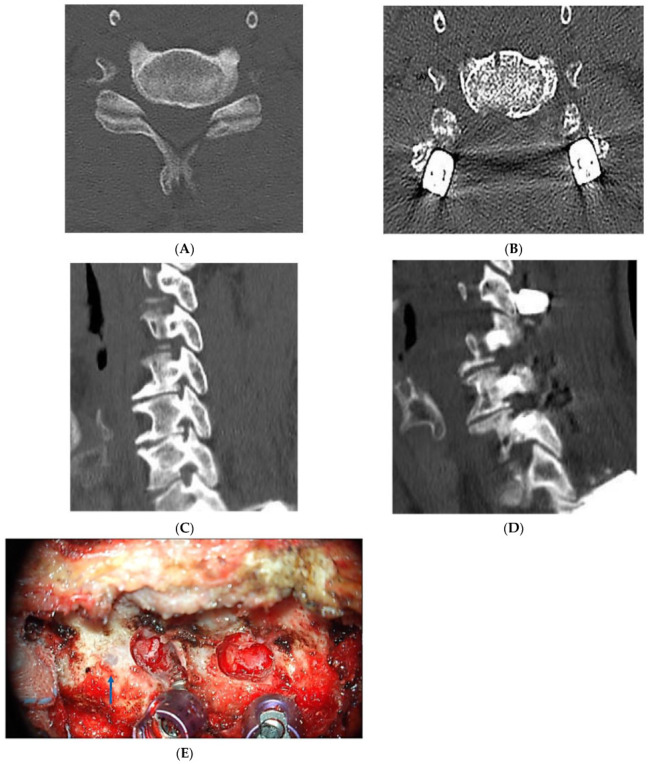
Posterior foraminotomy in the posterior pedicle group. (**A**) Preoperative CT axial image showing foraminal stenosis. (**B**) Postoperative axial image showing foraminotomy and laminectomy with pedicle screws. (**C**) Preoperative CT sagittal image showing foraminal stenosis. (**D**) Postoperative CT sagittal image showing extensive foraminotomy from the cephalic to caudal pedicles. (**E**) Intraoperative photographs showing foramen decompression and pedicle screws before and after expansive laminectomy. It is our surgical preference to perform foraminotomy before laminectomy. Arrow indicates a pre-drilled entry point for lateral mass screws. This figure is present to demonstrate the contrast regarding the amount of foraminotomy. When pedicle screws are used, expansive foraminotomy can be carried out from pedicle to pedicle. In contrast, when LMS is used, such expansive foraminotomy cannot be performed to preserve sufficient bone stock to secure the stability of lateral mass screw fixation, which has been proven by biomechanical study [30]. LMS, lateral mass screws; CT, computed tomography.

**Table 1 jcm-12-02227-t001:** Baseline demographic and clinical characteristics of the studied groups.

	PAP LMS (I) (n = 208)	PAP Pedicle (II) (n = 72)	Post. Pedicle (III) (n = 82)	*p*-Value
**Age/year**				
**Mean ± SD**	63.22 ± 10.6	65.52 ± 9.6	66.62 ± 12.1	**=0.037 ***
***p*-value ****	I vs. II = 0.122	II vs. III = 0.533	**I vs. III =0.017**	
**Sex**				**=0.005 *****
**Female**	53 (25.5%)	31 (43.1%)	18 (22%)	
**Male**	155 (74.5%)	41 (56.9%)	64 (78%)	
**Weight/kg**				
**Mean ± SD**	64.12 ± 5.8	64.23 ± 5.7	63.64 ± 5.2	=0.894 *
***p*-value ****	I vs. II = 0.992	II vs. III = 0.789	I vs. III = 0.824	
**Height/kg**				
**Mean ± SD**	159.11 ± 6.6	159.61 ± 6.3	158.63 ± 6.0	=0.833 *
***p*-value ****	I vs. II = 0.964	II vs. III = 0.624	I vs. III = 0.701	
**DM**	39 (18.8%)	10 (13.9%)	17 (20.7%)	=0.524 ***
**HTN**	69 (33.2%)	28 (38.9%)	29 (35.4%)	=0.675 ***
**Smoker**	69 (33.2%)	25 (34.7%)	30 (36.6%)	=0.855 ***
**Alcohol Abuse**	87 (41.8%)	31 (43.1%)	37 (45.1%)	=0.877 ***
**Osteoporosis**	62 (29.8%)	26 (36.1%)	25 (30.5%)	=0.602 ***
**Fused Levels**				
**Median (IQR)**	4 (0)	4 (0)	4 (2)	=0.560 ^$^
**C5 Palsy**	21 (10.1%)	3 (4.2%)	0 (0%)	**<0.001 ^#^**

* ANOVA test was used to compare the mean difference between groups. ** Post hoc test was used for pairwise comparison with Bonferroni correction. *** Chi-square test was used to compare the proportion difference between groups. ^$^ Independent samples Kruskal–Wallis test was used to compare the median difference between groups. ^#^ Monte Carlo exact test was used to compare the proportion difference between groups. IQR, interquartile range; LMS, lateral mass screws; max, maximum; min, minimum; PAP, posterior–anterior–posterior.

**Table 2 jcm-12-02227-t002:** Cervical lordosis parameters in the studied groups.

	PAP LMS (I) (n = 208)	PAP Pedicle (II) (n = 72)	Post. Pedicle (III) (n = 82)	*p*-Value
**Pre-C2 Slope (degrees)**				**=0.002 ***
**Median (IQR)**	14.5 (12)	17.5 (10)	15.5 (12)	
***p*-value ****	**I vs. II = 0.002**	**II vs. III = 0.008**	I vs. III = 0.789	
**Pre-T1 Slope (degrees)**				
**Median (IQR)**	23 (11)	17.5 (10)	25.5 (10)	**<0.001 ***
** *p* ** **-value ****	**I vs. II = 0.001**	**II vs. III < 0.001**	I vs. III = 0.063	
**Pre-C2–7 SVA (mm)**				
**Median (IQR)**	19 (17)	21 (9)	19.5 (13)	=0.809 *
**Pre-C2–7 Cobb Angle (degrees)**				
**Median (IQR)**	7 (14)	5.5 (9.5)	9 (15)	**=0.021 ***
***p*-value ****	I vs. II = 0.881	**II vs. III = 0.029**	I vs. III = 0.061	
**Pre-T1-CL (degrees)**				
**Median (IQR)**	16 (12)	14 (19)	15 (15)	**=0.031 ***
** *p* ** **-value ****	**I vs. II = 0.036**	II vs. III = 0.921	I vs. III = 0.444	
**Post-C2 Slope (degrees)**				
**Median (IQR)**	12.5 (12)	13 (7)	15.5 (9.5)	**=0.043 ***
** *p* ** **-value ****	I vs. II = 0.987	II vs. III = 0.454	**I vs. III = 0.036**	

* Independent samples Kruskal–Wallis test was used to compare the median difference between groups. ** Post hoc test was used for pairwise comparison with Bonferroni correction. IQR, interquartile range; LMS, lateral mass screws; max, maximum; min, minimum; PAP, posterior–anterior–posterior; SD, standard deviation; SVA, sagittal vertical axis; T1s-CL, T1 slope minus cervical lordosis (C2–C7 Cobb angle).

**Table 3 jcm-12-02227-t003:** Cervical lordosis parameters in the studied groups—continuation.

	PAP LMS (I) (n = 208)	PAP Pedicle (II) (n = 72)	Post. Pedicle (III) (n = 82)	*p*-Value
**Post-T1 Slope (degrees)**				**<0.001 ***
**Median (IQR)**	30 (10)	24 (10)	27 (12)	
***p*-value ****	**I vs. II = 0.001**	**II vs. III = 0.004**	I vs. III = 0.842	
**Post-C2–7 SVA (mm)**				
**Median (IQR)**	23.5 (16)	21.5 (9)	22 (10)	=0.322 *
**Post-C2–7 Cobb Angle (degrees)**				
**Median (IQR)**	18 (13)	26 (11)	13 (17)	**<0.001 ***
***p*-value ****	**I vs. II < 0.001**	**II vs. III < 0.001**	I vs. III = 0.055	
**Post-T1-CL (degrees)**				
**Median (IQR)**	13 (11)	2 (14)	12 (16)	**<0.001 ***
** *p* ** **-value ****	**I vs. II < 0.001**	**II vs. III < 0.001**	I vs. III = 0.745	
**Delta C2 (postoperative–preoperative) (degrees)**				
**Median (IQR)**	−3 (10)	−6 (11)	0.4 (7.5)	**<0.001 ***
** *p* ** **-value ****	I vs. II = 0.207	**II vs. III < 0.001**	**I vs. III = 0.003**	

* Independent samples Kruskal–Wallis test was used to compare the median difference between groups. ** Post hoc test was used for pairwise comparison with Bonferroni correction.

**Table 4 jcm-12-02227-t004:** NDI, JOA (pre- and postoperative results at 6 months and 1 year) in the studied groups.

	PAP LMS (I) (n = 208)	PAP Pedicle (II) (n = 72)	Post. Pedicle (III) (n = 82)	*p*-Value
**Preoperative JOA**				=0.256 *
**Median (IQR)**	12 (7)	11 (6)	11 (8)	
**JOA at 6 months post-operation**				=0.843 ***
**Mean ± SD**	15.0 ± 1.99	15.3 ± 1.70	15.2 ± 1.45	
**JOA at 1 year post-operation**				
**Mean ± SD**	15.5 ± 1.89	16.1 ± 1.48	16.5 ± 1.12	**=0.011 *****
***p*-value ****	**I vs. II = 0.024**	II vs. III = 0.229	**I vs. III = 0.008**	
**Preoperative NDI**				
**Median (IQR)**	20 (17)	19 (12)	20 (8)	**=0.047 ***
** *p* ** **-value ****	I vs. II = 0.333	**II vs. III = 0.040**	I vs. III = 0.489	
**NDI at 6 months post-operation**				=0.188 *
**Median (IQR)**	15 (11)	14 (13)	12 (11)	
**NDI at 1 year post-operation**				
**Median (IQR)**	13 (11)	12 (5.5)	12 (4)	**=0.046 ***
** *p* ** **-value ****	I vs. II = 0.183	II vs. III = 0.928	**I vs. III = 0.040**	

* Independent samples Kruskal–Wallis test was used to compare the median difference between groups. ** Post hoc test was used for pairwise comparison with Bonferroni correction. *** ANOVA test was used to compare the mean difference between groups. IQR, interquartile range; JOA, Japanese Orthopedic Association; LMS, lateral mass screws; max, maximum; min, minimum; NDI, Neck Disability Index; PAP, posterior–anterior–posterior; SD, standard deviation.

## Data Availability

The datasets generated for this study are available on request to the corresponding author. The data are not publicly available due to privacy reasons.

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
