# Peer review of "Posterior Preventive Foraminotomy before Laminectomy Combined with Pedicle Screw Fixation May Decrease the Incidence of C5 Palsy in Complex Cervical Spine Surgery in Patients with Severe Myeloradiculopathy"

_jcm, 2023, doi:10.3390/jcm12062227_

Round 1

Reviewer 1 Report (New Reviewer)

Aithors described about C5 palsy after cervical fusion surgery. The results were clear and informative. I want authors to change some minor hypo and discussion part.

1) Figure legends were written in capital (A), (B), (C), ,,,,altough figure captions were written in (a), (b), (c)....

2) Please add "the heat" as one of the reasons of C5 palsy after cervical surgery in discussion part. Please refer the following articles written by Takenaka and Suhrab,    

Takenaka S, Hosono N, Mukai Y, Tateishi K, Fuji T. Significant reduction in the incidence of C5 palsy after cervical laminoplasty using chilled irrigation water. Bone Joint J. 2016 Jan;98-B(1):117-24.

Rahmani MS, Terai H, Akhgar J, Suzuki A, Toyoda H, Hoshino M, Tamai K, Ahmadi SA, Hayashi K, Takahashi S, Nakamura H. Anatomical analysis of human ligamentum flavum in the cervical spine: Special consideration to the attachments, coverage, and lateral extent. J Orthop Sci. 2017 Nov;22(6):994-1000.

3) Please add clear indications for each surgical method. Is it decided by health care system or symptoms/ specific radiographic findings/ cervical alignment ? 

Author Response

Thank you for your meticulous review. Authors tried to do the best to meet the comments raised by the reviewers.

1) Figure capital has been modified

2) Additions have been added to the page 13 271-277, reflecting the contents of the comment.

‘According to Shurab Rachmani et al., ligamentum flavum does not enter the foramen unlike the lumba spine. Therefore, it is exposed to the risk of thermal damage during manipulation around the foramen. Takenaga et. al reported lower C5 palsy in the group using chilled irrigation than in the control group in 400 cases of open door laminoplasty [56,57]. In all cases of this study, during manipulation around the fora-men, To reduce the possibility of thermal injury that may occur during drilling, saline irrigation was performed.’

3) Applies to both. Surgical indication is text-based, but if the patients were in the gray zone of surgical indications, there is a risk of reduction in fees from the national health insurance. so it was decided very conservatively and protectively. All decisions are double checked by surgeons and referee system of the institute.

Reviewer 2 Report (New Reviewer)

la traduzione

Risultato di traduzionThe study contains some significant and interesting messages. The topic covered is also of practical interest.

The study contains some significant and interesting messages. The topic covered is also of practical interest. The study is retrospective and the identified groups have different numbers. However, the study suggests two important conclusions: 1) Performing a preventive posterior foraminotomy may reduce the risk of occurrence of postoperative C5 root lesion 2) The execution of foraminotomy associated with the use of pedicle screws allows greater correction in lordosis compared to the use of screws in the lateral masses The authors should better specify that this type of surgery is to be reserved for particularly serious cases. In the materials and methods, the description of the groups with the different surgical options should be somewhat simplified and made clearer by clearly indicating anterior and posterior approaches. The grading of foraminal stenosis needs to be better explained in particular to motivate the use of a lumbar grading method. The images in Figure 1 are adequate The images of figures 2 and 3 should be reduced and presented in a better graphic format. The intraoperative photographs in Figure 3 are of low quality and could be omitted. Figure 4 may be helpful. The results also need to be a little simplified and made more immediate. The tables are adequate. In the discussion the part concerning the review and refunds can be omitted. Overall, the work is interesting and the reported series is the first in the literature to make this type of comparison.

Author Response

 In the materials and methods, the description of the groups with the different surgical options should be somewhat simplified and made clearer by clearly indicating anterior and posterior approaches. 

- Thank you for your meticulous review. Authors tried to do the best to meet the comments raised by the reviewers.

Additions have been added to the page 2 81-83, reflecting the contents of the comment.

‘The anterior and posterior combined approach was performed only in cases with k-line negative and rigid kyphosis in lesions of 3 or more stages. [17]’

The grading of foraminal stenosis needs to be better explained in particular to motivate the use of a lumbar grading method. 

Additions have been added to the page 2 85-89, reflecting the contents of the comment.

The severity of foraminal stenosis could be graded as 1-2 of foraminal stenosis by the MRI Grading System for cervical foraminal stenosis from axial scan; grade 0 =absence of foraminal stenosis with the narrowest width of the neural foramen being greater than the width of the extraforaminal nerve root (EFNR); grade 1 = the narrowest width of the neural foramen equal to or greater than 50% of the width of the EFNR; and grade 2 = the narrowest width of the neural foramen equal to or less than 50% of the width of the EFNR [18,19]

Since the grading of cervical foraminal stenosis is based on MRI axial stenosis, in the case of severe upper and lower stenosis seen in the sagittal oblique view, although less severe in the axial view, stenosis in the perineural area was evaluated as in the lumbar spine foraminal stenosis grading system.

The images in Figure 1 are adequate The images of figures 2 and 3 should be reduced and presented in a better graphic format. The intraoperative photographs in Figure 3 are of low quality and could be omitted. Figure 4 may be helpful. 

Edited to reflect the comment

: intraoperative figure was omiited as you suggested.

The results also need to be a little simplified and made more immediate. The tables are adequate.

Thanks for your comments. Following your suggestion, the results were shortened and simplified by removal of redundancies.

In the discussion the part concerning the review and refunds can be omitted.

As previous mentioned in the responses to the reviewer 1’s comments, Surgical indication is text-based, but if the patients were in the gray zone of surgical indications, there is a risk of reduction in fees from the national health insurance. so it was decided very conservatively and protectively. All decisions are double checked by surgeons and referee system of the institute. This is the only and unique health system of authors’ country, which might be helpful to understand the surgical techniques and decision process of the patients.

Overall, the work is interesting and the reported series is the first in the literature to make this type of comparison.

Thank you for your positive comments!

This manuscript is a resubmission of an earlier submission. The following is a list of the peer review reports and author responses from that submission.

Round 1

Reviewer 1 Report

The scientific hypothesis presented in this paper is that foraminal size before surgery may be a risk factor for C5 nerve root palsy. The authors sought to answer this question by comparing the incidence of C5 nerve root palsy after three different surgical approaches involving foraminal decompression. The entire retrospective study did not account for the size of the intervertebral foramina before and after surgery in different groups, and the study design could not answer the scientific questions raised by the researchers. There are clear flaws in the study design.